

# Simulation of the Performance and Scalability of MPI Communications of Atmospheric Models running on Exascale Supercomputers

Yongjun ZHENG [*1] and Philippe MARGUINAUD[1]

[1]*Centre National de Recherches Météorologiques, Météo France, Toulouse* 31057

## Abstract

In this study, we identify the key MPI operations required in atmospheric modelling; then, we use a skeleton program and a simulation framework (based on SST/macro simulation package) to simulate these MPI operations (transposition, halo exchange, and allreduce), with the perspective of future exascale machines in mind. The experimental results show that the choice of the collective algorithm has a great impact on the performance of communications, in particular we find that the generalized ring-k algorithm for the alltoallv operation and the generalized recursive-k algorithm for the allreduce operation perform the best. In addition, we observe that the impacts of interconnect topologies and routing algorithms on the performance and scalability of transpositions, halo exchange, and allreduce operations are significant, however, that the routing algorithm has a negligible impact on the performance of allreduce operations because of its small message size. It is impossible to infinitely grow bandwidth and reduce latency due to hardware limitations, thus, congestion may occur and limit the continuous improvement of the performance of communications. The experiments show that the performance of communications can be improved when congestion is mitigated by a proper configuration of the topology and routing algorithm, which uniformly distribute the congestion over

---
*Corresponding author: yongjun.zheng@meteo.fr





the interconnect network to avoid the hotspots and bottlenecks caused by congestion. It

is generally believed that the transpositions seriously limit the scalability of the spectral

models. The experiments show that although the communication time of the transposi-

tion is larger than those of the wide halo exchange for the Semi-Lagrangian method and

the allreduce in the GCR iterative solver for the Semi-Implicit method below $2 \times 10^5$ MPI

processes, the transposition whose communication time decreases quickly as the number

of MPI processes increases demonstrates strong scalability in the case of very large grids

and moderate latencies; the halo exchange whose communication time decreases more

slowly than that of transposition as the number of MPI processes increases reveals its

weak scalability; in contrast, the allreduce whose communication time increases as the

number of MPI processes increases does not scale well. From this point of view, the scal-

ability of the spectral models could still be acceptable, therefore it seems to be premature

to conclude that the scalability of the grid-point models is better than that of spectral

models at exascale, unless innovative methods are exploited to mitigate the problem of

the scalability presented in the grid-point models.

**Keyword**: performance, scalability, MPI, communication, transposition, halo exchange,

all reduce, topology, routing, bandwidth, latency

# 1   Introduction

Current high performance computing (HPC) systems have thousands of nodes and millions

of cores. According to the 49th TOP500 list (`www.top500.org`) published on June 20, 2017,

the fastest machine (Sunway TaihuLight) had over than 10 million cores with a peak perfor-

mance approximately 125 PFlops (1 PFlops=$10^{15}$ floating-point operations per second), and

the second HPC (Tianhe-2) is made up of 16,000 nodes and has more than 3 million cores with

a peak performance approximately 55 PFlops. It is estimated that in the near future, HPC

systems will dramatically scale up in size. Next decade, it is envisaged that exascale HPC

system with millions of nodes and thousands of cores per node, whose peak performance ap-

proaches to or is beyond 1 EFlops (1 EFlops=$10^3$ PFlops), will become available (Engelmann,

2014; Lagadapati et al., 2016). Exascale HPC poses several challenges in terms of power con-

sumption, performance, scalability, programmability, and resilience. The interconnect net-

work of exascale HPC system becomes larger and more complex, and its performance which

largely determines the overall performance of the HPC system is crucial to the performance

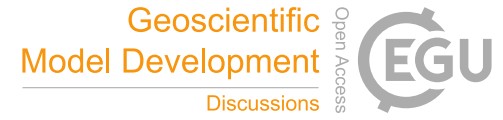

of distributed applications. Designing energy-efficient cost-scalable interconnect networks and communication-efficient scalable distributed applications is an important component of HPC hardware/software co-design to address these challenges. Thus, evaluating and predicting the communication behaviour of distributed applications is obligatory; it is only feasible by modelling the communications and the underlying interconnect network, especially for the future supercomputer.

Investigating the performance of distributed applications on future architectures and the impact of different architectures on the performance by simulation is a hardware/software co-design approach for paving the way to exascale HPCs. Analytical interconnect network simulation based on an analytical conceptual model is fast and scalable, but comes at the cost of accuracy owing to its unrealistic simplification (Hoefler et al., 2010). Discrete event simulation (DES) is often used to simulate the interconnect network, and it provides high fidelity since the communication is simulated in more detailed level (e.g., flit, packet, or flow levels) to take into account congestion (Janssen et al., 2010; Böhm and Engelmann, 2011; Dechev and Ahn, 2013; Acun et al., 2015; Jain et al., 2016; Wolfe et al., 2016; Degomme et al., 2017; Mubarak et al., 2017). Sequential DES lacks scalability owing to its large memory footprints and long execution time (Degomme et al., 2017). Parallel DES (PDES) is scalable since it can reduce the memory required per node, but its parallel efficiency is not very good because of frequent global synchronization of conservative PDES (Janssen et al., 2010) or high rollback overhead of optimistic PDES (Acun et al., 2015; Jain et al., 2016; Wolfe et al., 2016). Generally, the simulation of distributed applications can be divided into two complementary categories: offline and online simulations. Offline simulation replays the communication traces from the application running on a current HPC system. It is sufficient to understand the performance and discover the bottleneck of full distributed applications on the available HPC system (Tikir et al., 2009; Noeth et al., 2009; Núñez et al., 2010; Dechev and Ahn, 2013; Casanova et al., 2015; Acun et al., 2015; Jain et al., 2016; Lagadapati et al., 2016); however, is not very scalable because of the huge traces for numerous processes and limited extrapolation to future architecture (Hoefler et al., 2010; Núñez et al., 2010). Online simulation has full scalability to future system by running the skeleton program on the top of simulators (Zheng et al., 2004; Janssen et al., 2010; Engelmann, 2014; Degomme et al., 2017), but has the challenge of developing a skeleton program from a complex distributed application. Most simulations in the aforementioned literatures have demonstrated the scalability of simulators. The simulator xSim (Engelmann,

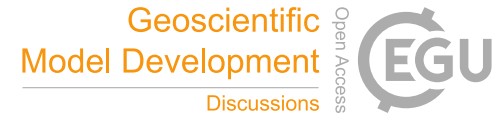


2014) simulated a very simple MPI program, which only calls MPI_Init and MPI_Finalize without any communication and computation, up to $2^{27}$ processes. For collective MPI operations, Hoefler et al. (2010) obtained an MPI_Allreduce simulation of 8 million processes without consideration of congestion using LogGOPSim, Engelmann (2014) achieved an MPI_Reduce simulation of $2^{24}$ processes, and Degomme et al. (2017) demonstrated an MPI_Allreduce simulation of 65536 processes using SimGrid. For simulations at application level, Jain et al. (2016) used the TraceR simulator based on CODES and ROSS to replay $4.6 \times 10^4$ process traces of several communication patterns that are used in a wide range of applications. In addition, Mubarak et al. (2017) presented a $1.1 \times 10^5$ process simulations of two multigrid applications. However, to the best of our knowledge, there is no exascale simulation of complex communication patterns such as the MPI transposition (Multiple simultaneous MPI_Alltoallv) for the spectral method and the wide halo exchange (the width of a halo may be greater than the subdomain size of its direct neighbours) for the Semi-Lagrangian method used in atmospheric models.

With the rapid development of increasingly powerful supercomputers in recent years, numerical weather prediction (NWP) models have increasingly sophisticated physical and dynamical processes, and their resolution is getting higher and higher. Nowadays, the horizontal resolution of global NWP model is in the order of 10 kilometres. Many operational global spectral NWP models such as IFS at ECMWF, ARPEGE at METEO-FRANCE, and GFS at NCEP are based on the spherical harmonics transform method that includes Fourier transforms in the zonal direction and Legendre transforms in the meridional direction (Ehrendorfer, 2012). Moreover, some regional spectral models such as AROME at METEO-FRANCE (Seity et al., 2011) and RSM at NCEP (Juang et al., 1997) use the Bi-Fourier transform method. The Fourier transforms can be computed efficiently by fast Fourier transform (FFT) (Temperton, 1983). Even with the introduction of fast Legendre transform (FLT) to reduce the growing computational cost of increasing resolution of global spectral models (Wedi et al., 2013), it is believed that global spectral method is prohibitively expensive for very high resolution (Wedi, 2014).

A global (regional) spectral model performs FFT and FLT (FFT) in the zonal direction and the meridional direction, respectively. Because both transforms require all values in the corresponding directions, the parallelization of spectral method in global (regional) model is usually conducted to exploit the horizontal domain decomposition only in the zonal direction and meridional directions for FFT and FLT (FFT), respectively (Barros et al., 1995; Kanamitsu et al., 2005). Owing to the horizontal domain decomposition in a single horizontal direction for the



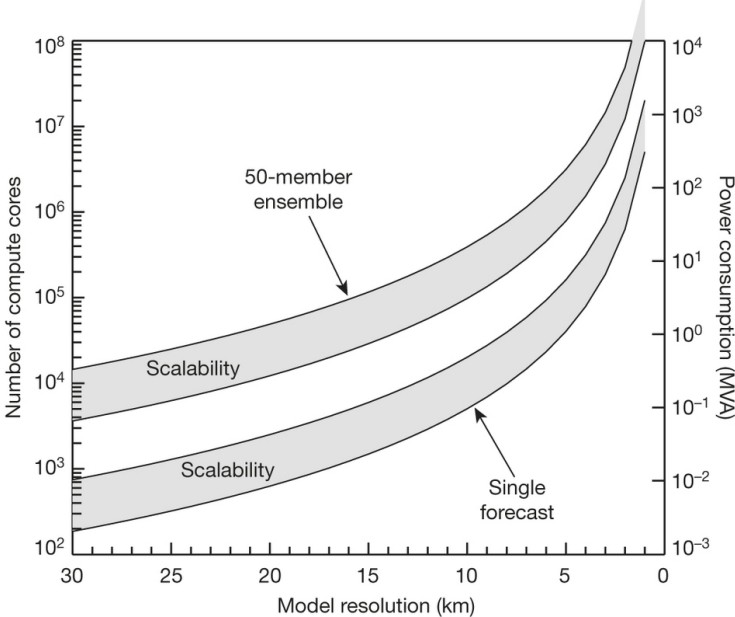

Fig. 1: CPU and power requirements as a function of NWP model resolution, adapted from Bauer et al. (2015). The left and right y axes are the number of cores and the power (in megavolt amps), respectively, required for a single 10-day model forecast (the lower shaded area including its bounds) and a 50-member ensemble forecast (the upper shaded area including its bounds) as a function of model resolution, respectively, based on current model code and compute technology. The lower and upper bounds of each shaded area indicate perfect scaling and inefficient scaling, respectively.

parallelization of spectral transforms, there is a transposition between the spectral transforms in the zonal direction and meridional directions. MPI (Message Passing Interface) transposition is an all-to-all personalized communication which can cause significant congestion over interconnect network when the number of MPI tasks and the amount of exchanged data are large, and results in severe communication delay. Bauer et al. (2015) estimated that a global NWP model with a two-kilometre horizontal resolution requires one million compute cores for a single 10-day forecast (Fig. 1). With one million compute cores, the performance and scalability of the MPI transposition become of paramount importance for a high resolution global spectral model. Thus, evaluating and predicting the performance and scalability of MPI transposition at exascale is one of the foremost subjects of this study.

The Semi-Lagrangian (SL) method is a highly efficient technique for the transport of momentum, heat and mass in the NWP model because of its unconditional stability which permits a long time step (Staniforth and Côté, 1991; Hortal, 2002). However, it is known that the MPI

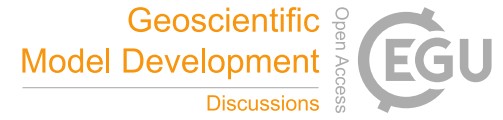



exchange of wide halo required for the interpolation at the departure point of high wind-speed particles near the boundary of the subdomain causes significant communication overhead as resolution increases towards kilometres scale and the HPC systems move towards exascale. This communication overhead could reduce the efficiency of the SL method; thus, modelling the performance and scalability of wide halo exchange at exascale is essential and is another subject of this study.

With consideration of the efficiency of the Legendre transform and the scalability of MPI transposition that may arise in the global spectral model on exascale HPC systems, a couple of global grid-point models have recently been developed (Lin, 2004; Satoh et al., 2008; Qaddouri and Lee, 2011; Skamarock et al., 2012; Dubos et al., 2015; Zangl et al., 2015; Kuhnlein and Smol, 2017). Since spherical harmonics are eigenfunctions of the Helmholtz operator, the Semi-Implicit (SI) method is usually adopted in order to implicitly handle the fast waves in the global spectral model to allow stable integration with a large time step (Robert et al., 1972; Hoskins and Simmons, 1975). However, for a grid-point model, the three-dimensional Helmholtz equation is usually solved using Krylov subspace methods such as the generalized conjugate residual (GCR) method (Eisenstat et al., 1983), and a global synchronization for the inner product in Krylov subspace methods may become the bottleneck at exascale (Li et al., 2013; Sanan et al., 2016). As it is not clear whether the three-dimensional Helmholtz equation can be solved efficiently in a scalable manner, most of the aforementioned models use a horizontally explicit vertically implicit (HEVI) scheme. The HEVI scheme typically requires some damping for numerical stability (Satoh et al., 2008; Skamarock et al., 2012; Zangl et al., 2015), and its time step is smaller than that of the SI method (Sandbach et al., 2015). Therefore, it is desirable to know whether the SI method is viable or even advantageous for very high resolution grid-point models running on exascale HPC systems. Thus, it is valuable to explore the performance and scalability of global synchronization in solving the three-dimensional Helmholtz equation using Krylov subspace methods; this forms the third subject of this study.

In this paper, we present the application of SST/macro 7.1, a coarse-grained parallel discrete event simulator, to investigate the communication performance and scalability of atmospheric models for future exascale supercomputers. The remainder of the paper is organized as follows. Section 2 introduces the simulation environment, the SST/macro simulator, and our optimizations for reducing the memory footprint and accelerating the simulations. Section 3 reviews three key MPI operations used in the atmospheric models. Section 4 presents and





analyses the experimental results of the modelling communication of the atmospheric model using SST/macro. Finally, we summarize the conclusions and discuss future work in section 5.

## 2 Simulation Environment

### 2.1 Parallel Discrete Event Simulation

Modelling application performance on exascale HPC systems with millions of nodes and a complex interconnect network requires that the simulation can be decomposed into small tasks that efficiently run in parallel to overcome the problem of large memory footprint and long simulation time. PDES is such an approach for exascale simulation. Each worker in PDES is a logical process (LP) that models a specific component such as a node, a switch, or an MPI process of the simulated MPI application. These LPs are mapped to the physical processing elements (PEs) that actually run the simulator. An event is an action such as sending an MPI message or executing a computation between consecutive communications. Each event has its start and stop times, so the events must be processed without violating their time ordering. To model the performance of an application, PDES captures time duration and advances the virtual time of the application by sending timestamped events between LPs.

PDES usually adopts conservative or optimistic parallelized strategies. The conservative approach maintains the time ordering of events by synchronization to guarantee that no early events arrive after the current event. Frequent synchronization is time-consuming so the efficiency of the conservative approach is highly dependent on the look ahead time; a larger look ahead time (that means less synchronization) allows a much greater parallelism. The optimistic approach allows LPs to run events at the risk of time-ordering violations. Events must be rolled back when time-ordering violations occurs. Rollback not only induces significant overhead, but also requires extra storage for the event list. Rollback presents special challenges for online simulation, so SST/macro adopts a conservative approach (Wike and Kenny, 2014).

### 2.2 SST/macro Simulator

Considering that the offline trace-driven simulation does not provide an easy way for extrapolating to future architectures, the online simulator SST/macro is selected here to model the communications of the atmospheric models for future exascale HPC systems. SST/macro is a





coarse-grained parallel discrete event simulator which provides the best cost/accuracy trade-off simulation for large-scale distributed applications (Janssen et al., 2010). SST/macro is driven by either a trace file or a skeleton application. A skeleton application can be constructed from scratch, or from an existing application manually or automatically by source-to-source translation tools. SST/macro intercepts the communications issued from the skeleton program to estimate their time rather than actually execute it by linking the skeleton application to the SST/macro library instead of the real MPI library. Since the purpose of this study is to investigate the performance and scalability of communications in an atmospheric model, we construct the communication-only skeleton program from scratch by identifying the key MPI operations taking place in the atmospheric models.

Congestion is a significant factor that affects the performance and scalability of MPI applications running on exascale HPC systems. SST/macro has three network models: the analytical model transfers the whole message over the network from point-to-point without packetizing and estimates the time delay $\Delta t$ predominantly based on the logP approximation

$$\Delta t = \alpha + \beta N, \tag{1}$$

where $\alpha$ is the communication latency, $\beta$ is the inverse bandwidth in second per byte, and $N$ is the message size in bytes; the packet-level model PISCES (Packet-flow Interconnect Simulation for Congestion at Extreme Scale) divides the message into packets and transfers the packets individually; the flow-level model will be deprecated in the future. Compared to the SimGrid simulator, the packet-level model of SST/macro produces almost identical results (figure omitted). Acun et al. (2015) also found that the SST/macro online simulation is very similar to the TraceR simulation. Thus, we adopt the PISCES model with a cut-through mechanism (SNL, 2017) to better account for the congestion. SST/macro provides three abstract machine models for nodes: the AMM1 model is the simplest one which grants exclusive access to the memory, the AMM2 model allows multiple CPUs or NICs (network interface controller) to share the memory bandwidth by defining the maximum memory bandwidth allocated for each component, the AMM3 model goes one further step to distinguish between the network link bandwidth and the switch bandwidth. In this paper, the AMM1 model with one single-core CPU per node is adopted since simulation of communications is the primary goal.

SST/macro provides several topologies of the interconnect network. In this study, three



types of topologies (Fig. 2) commonly used in current supercomputers, and their configurations
are investigated. Torus topology has been used in many supercomputers (Ajima et al., 2009).
In the torus network, messages hop along each dimension using taddthe shortest path routing
from the source to the destination (Fig. 2a), and its bisection bandwidth typically increases with
increasing dimension size of the torus topology. The practical implementation of the fattree
topology is an upside-down tree that typically employs all uniform commodity switches to
provide high bandwidth at higher levels by grouping corresponding switches of the same colour
(Fig. 2b). Fattree topology is widely adopted by many supercomputers for its scalability and
high path diversity (Leiserson, 1985); it usually uses a D-mod-k routing algorithm (Zahavi et al.,
2010) for desirable performance. A dragonfly network is a multi-level dense structure of which
the high-radix routers are connected in a dense even all-to-all manner at each level (Kim et al.,
2008). As shown in Fig. 2c, a typical dragonfly network consists of two levels: the routers at
the first level are divided into groups and routers in each group form a two-dimension mesh
of which each dimension is an all-to-all connected network; at the second level, the groups as
virtual routers are connected in an all-to-all manner (Alverson et al., 2015). There are three
available routing algorithms for dragonfly topology in SST/macro:

**minimal** transfers messages by the shortest path from the source to the destination. For
example, messages travel from the blue router in group 0 to the red router in group 2 via
the bottom-right corner in group 0 and the bottom-left corner in group 2 (Fig. 2c).

**valiant** randomly picks an intermediate router, and then uses a minimal routing algorithm to
transfer messages from the source to the intermediate router and from the intermediate
router to the destination. For example, the arrow path from the blue router in group 0
to the red router in group 2 goes via the intermediate yellow node in group 1 in Fig. 2c.

**ugal** checks the congestion, and either switches to the valiant routing algorithm if congestion
is too heavy, or otherwise uses the minimal routing algorithm.

Table 1 summaries the network topology configurations used in this paper. Torus-M (torus-
L) configuration is a 3D torus of 25x25x25 (75x25x25) size. Fattree-M (fattree-L) configuration
has 4 layers: the last layer consists of nodes while the other layers consist of switches with 25 (33)
descendant ports per switch. We tested four configurations of dragonfly topology. Dragonfly-
MM configuration has a medium size of a group of a 25x25 mesh with 25 nodes per switch





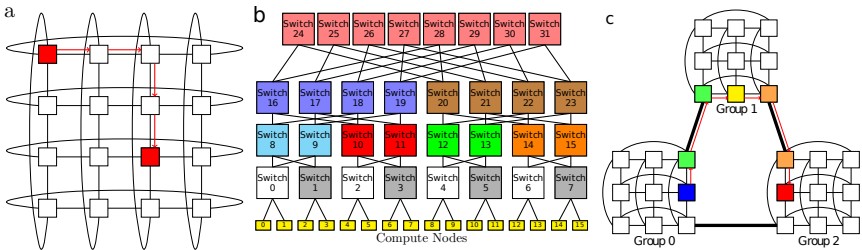

Fig. 2: Topology illustration: a, b, and c are the torus, fattree, and dragonfly topologies, respectively. Adapted from SNL (2017)

Table 1: Summary of the network topologies: the geometry of a torus topology specifies the size of each dimension; the first and second number in the geometry of a fattree topology are the number of layers and descendant ports per switch, respectively; the first two numbers and the last number in the geometry of a dragonfly topology indicate the group mesh size and the number of groups, respectively.

| name | geometry | switches | nodes per switch | nodes | radix |
|------|----------|----------|------------------|-------|-------|
| torus-M | 25,25,25 | 15625 | 25 | 390625 | 31 |
| fattree-M | 4,25 | 46875 | 25 | 390625 | 50 |
| dragonfly-MM | 25,25,25 | 15625 | 25 | 390625 | 97 |
| dragonfly-SL | 25,25,125 | 15625 | 5 | 390625 | 177 |
| dragonfly-LS | 125,125,5 | 15625 | 5 | 390625 | 257 |
| torus-L | 75,25,25 | 46875 | 25 | 1171875 | 31 |
| fattree-L | 4,33 | 107811 | 33 | 1185921 | 66 |
| dragonfly-ML | 25,25,75 | 46875 | 25 | 1171875 | 147 |

and medium number (=25) of groups. Dragonfly-SL configuration has a small size of a group

of a 25x25 mesh with 5 nodes per switch and large number (=125) of groups. Dragonfly-LS

configuration has a large size of a group of a 125x125 mesh with 5 nodes per switch and small

number (=5) of groups. Dragonfly-ML configuration has a medium size of a group of a 25x25

mesh with 25 nodes per switch and large number (=75) of groups. The fattree configuration

has a significant larger number of switches than other topologies for the same number of nodes,

which indicates that fattree is not cost- or energy-efficient. All the configurations with 390625

nodes are used for simulating transposition for the spectral transform method. Torus-L, fattree-

254 L, and dragonfly-ML with more than one million nodes are used for the cases of halo exchange

and allreduce communication since we cannot finish the simulation of transposition for the

spectral transform method (multiple simultaneous all-to-all personalized communications) on

such large configuration within 24 hours (see Section 3 for three key MPI communications in

the atmospheric model).



### 2.3   Reduce the Memory Footprint and Accelerate the Simulation

Although SST/macro is a parallel discrete event simulator that can reduce the memory footprint per node, its parallel efficiency degrades if more cores are used. Even with an MPI transposition of $10^5$ processes, this all-to-all personalized communication has almost $10^{10}$ discrete events, which consumes a considerable amount of memory and takes a very long time for simulation. Furthermore, almost every MPI program has a setup step to allocate memory for storing the setup information such as the parameters and the domain decomposition of all processes what each process must know in order to properly communicate with other processes, therefore, it needs to broadcast the parameters to and synchronize with all processes before actual communications and computation. Even if the setup information for a single process needs only $10^2$ bytes memory, a simulation of $10^5$ processes MPI transposition will need one terabyte ($10^2 \times 10^5 \times 10^5 = 10^{12}$ bytes) memory, which is not easily available on current computers if the simulator runs on a single node. In addition, the MPI operations in the setup step not only are time-consuming, but also affect subsequent communications. A common way to eliminate this effect is to iterate many times to obtain a robust estimation of communication time; however, one iteration is already very time-consuming for simulation. To circumvent the issue of setup steps, we use an external auxiliary program to create a shared memory segment on each node running SST/macro and initialize this memory with the setup information of all the simulated MPI processes. Then, we modified SST/macro to create a global variable and attach the shared memory to this global variable; this method not only reduces the memory footprint and eliminates the side effect of communications in the setup step, but also avoids the problem of filling up the memory address space if each simulated process attaches to the shared memory.

Large-scale application needs a large amount of memory for computation; and in some cases, such as spectral model, the whole memory for computation is exchanged between all the processes. Even when computation is not considered, a large amount of memory for the message buffers is usually required for MPI communications. Fortunately, the simulator only needs message size, the source/destination, and the message tag to model the communication; thus, it is not necessary to allocate actual memory. Since SST/macro can operate with null buffers, the message buffer is set to null in the skeleton application, which significantly reduces the size of memory required by the simulation of communication of the high resolution atmospheric



model.

## 3   Key MPI Operations in Atmospheric Models

### 3.1   Transposition for the Spectral Transform Method

A global spectral model generally uses spherical harmonics transform on the horizontal with

triangular truncation. The backward spherical harmonics transform is

$$f(\theta, \lambda) = \sum_{m=-M}^{M} \left( e^{im\lambda} \sum_{n=|m|}^{M} f_n^m P_n^m(\cos\theta) \right), \tag{2}$$

where $\theta$ and $\lambda$ are the colatitude and longitude, $f_n^m$ is the spectral coefficients of the field $f$, and

$P_n^m$ is the associated Legendre polynomials of degree $m$ and order $n$. Moreover, the forward

spherical harmonics transform is

$$f_n^m = \frac{1}{2} \int_{-1}^{1} \left( P_n^m(\cos\theta) \frac{1}{2\pi} \int_0^{2\pi} f(\theta, \lambda) e^{-im\lambda} d\lambda \right) d\cos\theta, \tag{3}$$

In (2), the backward Legendre transform of each $m$ can be computed independently; then,

the same is for the backward Fourier transform of each $\theta$. Similar to (3), the forward Fourier

transform of each $\theta$ can be computed independently; then, the same is for the forward Legendre

transform of each $m$. This leads to a natural way to parallelize the spectral transforms. If

we start with the grid-point space (Fig. 3a), which is decomposed by $cx/cy$ cores in the x/y

direction, $cy$ simultaneous xz slab MPI transpositions lead to the partition (Fig. 3b) with $cy/cx$

cores in the y/z direction, and a spectral transform such as a forward FFT can be performed

in parallel since data w.r.t. $\lambda$ are local to each core. Then, $cx$ simultaneous xy slab MPI

transpositions lead to the partition (Fig. 3c) with $cy/cx$ in the x/z direction, and a spectral

transform such as a forward FLT can be computed in parallel because data w.r.t. $\theta$ are now

local to each core. Finally, $cy$ simultaneous yz slab MPI transpositions lead to the spectral space

(Fig. 3d) with $cy/cx$ cores in the x/y direction, where the Semi-Implicit scheme can be easily

computed because spectral coefficients belonging to the same column are now local to the same

core. The backward transform is similar. It is of paramount importance that the partition of

the four stages described in Fig. 3 must be consistent so that multiple slab MPI transpositions

can be conducted simultaneously, which significantly reduces the communication time of MPI





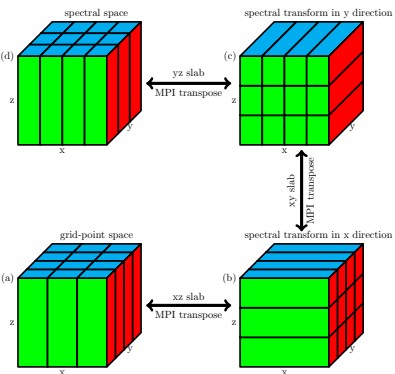

Fig. 3: Parallel scheme of regional spectral model: (a) 2D decomposition of 3D grid field with $cx/cy$ cores in the x/y direction, (b) 2D decomposition of 3D grid field with $cy/cx$ cores in the y/z direction , (c) 2D decomposition of 3D grid field with $cy/cx$ cores in the x/z direction, and (d) 2D decomposition of 3D grid field with $cy/cx$ cores in the x/y direction. Transposition between (a) and (b) can be conducted by $cy$ independent xz slab MPI transpositions, transposition between (b) and (c) can be conducted by $cx$ independent xy slab MPI transpositions, and transposition between (c) and (d) can be conducted by $cy$ independent yz slab MPI transpositions.

transpositions from one stage to another. It is worth noting that the number of grid points in one direction is not always a multiple of the number of cores in the corresponding direction; thus, the partition shown in Fig. 3 can use as many as possible computed cores without any limit on $cx$ or $cy$ provided $cx \times cy = ncpu$, and $cx$ or $cy$ is not greater than the number of grid points in the corresponding direction. It is generally believed that the MPI transpositions from one stage to another poses a great challenge to the scalability of spectral models because each slab MPI transposition is an all-to-all personalized communications which is the most complex and time-consuming all-to-all communication.

There are different algorithms for all-to-all personalized communication. Table 2 lists the three algorithms for all-to-all personalized communication, whose performance and scalability are investigated in this study. Algorithm ring-k is our proposal algorithm for all-to-all personalized communication which is a generalized ring alltoallv algorithm. In algorithm ring-k, each process communicates with $2k$ processes to reduce the stages of communications and make efficient use of the available bandwidth, and thus reduces the total communication time.





Table 2: Three algorithms for all-to-all personalized communication.

| name | description | stages |
|---|---|---|
| **burst** | Each process communicates with all other processes simultaneously by posting all non-block send and receive operations simultaneously. The burst messages cause significant congestion on the network. This algorithm is equivalent to the algorithm ring-k when k=n-1. | 1 |
| **bruck** | This algorithm is better for small message and a large latency since it has only $\lceil \log_2(n) \rceil$ stages of communications (Thakur et al., 2005). For $k^{th}$ stage, each process sends the messages whose destination process id has one at the $k^{th}$ bit (begin at Least Significant Bit) to process $i + 2^k$. | $\lceil \log_2(n) \rceil$ |
| **ring-k** | In the first stage, process $i$ sends to $i+1, \cdots, i+k$ and receive from $i-1, \cdots, i-k$ in a ring way (black arrows in Fig. 4a); in the second stage, process $i$ sends to $i+1+k, \cdots, i+2k$ and receive from $i-1-k, \cdots, i-2k$ in a ring way (blue arrows in Fig. 4a); this continues until all partners have been communicated with. This algorithm is a generalization of the ring algorithm and efficiently uses the available bandwidth by proper selection of radix $k$. | $\lceil \frac{n-1}{k} \rceil$ |

### 3.2 Halo Exchange for Semi-Lagrangian Method

The SL method solves the transport equation:

$$\frac{D\phi}{Dt} = \frac{\partial \phi}{\partial t} + u\frac{\partial \phi}{\partial x} + v\frac{\partial \phi}{\partial y} + w\frac{\partial \phi}{\partial z} = 0, \tag{4}$$

where the scalar field $\phi$ is advected by the 3D wind $\mathbf{V} = (u, v, w)$. In the SL method, the

grid-point value of the scalar field $\phi$ at next time step $t + \Delta t$ can be found by integrating (4)

along the trajectory of the fluid parcel (Staniforth and Côté, 1991; Hortal, 2002)

$$\int_t^{t+\Delta t} \frac{D\phi}{Dt} dt = 0 \rightarrow \phi^{t+\Delta t} = \phi_d^t, \tag{5}$$

where $\phi^{t+\Delta t}$ is the value of the fluid parcel $\phi$ arriving at any grid point at $t + \Delta t$, and $\phi_d^t$ is the

value of the same fluid parcel at its departure point $d$ and departure time $t$. This means that

the value of the scalar field $\phi$ at any grid point at $t + \Delta t$ is equal to its value at the departure

point $d$ and the departure time $t$. The departure point $d$ usually does not coincide with any grid

point, so the value of $\phi_d^t$ is obtained by interpolation using the surrounding grid-point values $\phi^t$

at time $t$. The departure point $d$ is determined by iteratively solving the trajectory equation





(Staniforth and Côté, 1991; Hortal, 2002)

$$\frac{D\mathbf{r}}{Dt} = \mathbf{V}(\mathbf{r}, t) \rightarrow \mathbf{r}^{t+\Delta} - \mathbf{r}_d^t = \int_t^{t+\Delta t} \mathbf{V}(\mathbf{r}, t)dt, \tag{6}$$

where $\mathbf{r}^{t+\Delta t}$ and $\mathbf{r}_d^t$ are the position of the arrival and the departure point, respectively. From (6), it is obvious that the departure point is far from its arrival point if the wind speed is large. Thus, the departure point of one fluid parcel at the boundary of the subdomain of an MPI task is far from its boundary if the wind speed is large and the wind blows from the outside. To facilitate calculation of the departure point and its interpolation, MPI parallelization adopts a "maximum wind" halo approach so that the halo is sufficiently large for each MPI task to perform its SL calculations in parallel after exchanging the halo. This "maximum wind" halo is named "wide halo" since its width is significantly larger than that of the thin halo of finite difference methods whose stencils have compact support. With numerous MPI tasks, the width of a wide halo may be larger than the subdomain size of its direct neighbour, which implies that the process needs to exchange the halo with its neighbours and its neighbours' neighbours, which may result in a significant communication overhead which counteracts the efficiency of the favourite SL method, and pose a great challenge to the scalability of the SL method.

Fig. 4b demonstrates the halo exchange algorithm adopted in this paper. First, the algorithm posts the MPI non-block send and receive operations 1-4 simultaneously for the x-direction sweep. After the x-direction sweep, a y-direction sweep is performed in a similar way but the length of halo is extended to include the left and right haloes in the x-direction so that the four corners are exchanged properly. This algorithm needs two stages communications, but is simple to implement, especially for the wide halo exchange owing to its fixed regular communication pattern (Fig. 9d). In Fig. 9d, the pixels (near purple colour) tightly attached to the diagonal are due to the exchange in x-direction, the pixels of the same colour but off diagonal are due because of the periodicity in x-direction; the pixels (near orange or red colour) off diagonal are due to the exchange in y-direction, and the pixels of the same colour but far off diagonal are because of the periodicity in y-direction. This algorithm also applies to the thin halo exchange for finite difference methods which is extensively used in the grid-point models. The study emphasizes on the wide halo exchange, but the thin halo exchange is also investigated for comparison (see the red line in Fig. 9a).



### 3.3  Allreduce in Krylov Subspace Methods for the Semi-Implicit Method

The three-dimensional SI method leads to a large linear system which can be solved by Krylov subspace methods:

$$\mathbf{A}\mathbf{x} = \mathbf{b}, \qquad (7)$$

where $\mathbf{A}$ is a non-symmetric sparse matrix. Krylov subspace methods find the approximation $\mathbf{x}$ iteratively in a $k$-dimensional Krylov subspace:

$$\mathcal{K} = span(\mathbf{r}, \mathbf{A}\mathbf{r}, \mathbf{A}^2\mathbf{r}, \cdots, \mathbf{A}^{k-1}\mathbf{r}), \qquad (8)$$

where $\mathbf{r} = \mathbf{b} - \mathbf{A}\mathbf{x}$. To accelerate the convergence, preconditioning is generally used:

$$\mathbf{M}^{-1}\mathbf{A}\mathbf{x} = \mathbf{M}^{-1}\mathbf{b} \qquad (9)$$

where $\mathbf{M}$ approximates $\mathbf{A}$ well so that $\mathbf{M}^{-1}\mathbf{A}$ be conditioned better than $\mathbf{A}$ and $\mathbf{M}^{-1}$ can be computed cheaply. The GCR method is a Krylov subspace method of easy implementation and can be used with variable preconditioners. Algorithm 1 of GCR shows that there are two allreduces operations using the sum operation for the inner product in each iteration, thus, it has 2N allreduce operations if the GCR iterative solver reaches convergence in N iterations. Allreduce is an all-to-all communication and becomes expensive when the number of iterations becomes larger in GCR solver with numerous MPI processes.

Fig. 4c demonstrates the recursive-k algorithm for the allreduce operation, which is a generalization of the recursive doubling algorithm. Let $p = \lfloor \log_k(ncpu) \rfloor$, this algorithm has $2 + p$ stages of communications if the number of processes is not a power of radix k. In the first stage with stage id $j = 0$ (the first row in Fig. 4c), each remaining process whose id $i \notin [0, k^p - 1]$ sends its data to process $i - (ncpu - k^p)$ for the reduce operation. For the stage of stage id $j \in [1, p]$ (rows between the first row and second last row in Fig. 4c), each process whose id $i \in [0, k^p - 1]$ only reduces with the processes that are a distance of $k^{j-1}$ apart from itself. In the final stage with stage id $j = 1 + p$ (the second last row in Fig. 4c), each process whose id $i \notin [0, k^p - 1]$ receives its final result from process $i - (ncpu - k^p)$. The recursive-k algorithm uses large radix k to reduce the stages of communications and the overall communication time.





---

**Algorithm 1** Preconditioned GCR returns the solution $\mathbf{x}_i$ when convergence occurs where $\mathbf{x}_0$ is the first guess solution and $k$ is the number of iterations for restart.

---

1: **procedure** GCR($\mathbf{A}, \mathbf{M}, \mathbf{b}, \mathbf{x}_0, k$)
2: $\quad \mathbf{r}_0 \leftarrow \mathbf{b} - \mathbf{A}\mathbf{x}_0$
3: $\quad \mathbf{u}_0 \leftarrow \mathbf{M}^{-1}\mathbf{r}_0$
4: $\quad \mathbf{p}_0 \leftarrow \mathbf{u}_0$
5: $\quad \mathbf{s}_0 \leftarrow \mathbf{A}\mathbf{p}_0$
6: $\quad \gamma_0 \leftarrow <\mathbf{u}_0, \mathbf{s}_0>, \eta_0 \leftarrow <\mathbf{s}_0, \mathbf{s}_0>$ $\qquad\qquad\qquad$ ▷ Allreduce(sum) of two doubles
7: $\quad \alpha_0 \leftarrow \frac{\gamma_0}{\eta_0}$
8: $\quad$ **for** $i = 1, \cdots,$ until convergence **do**
9: $\quad\quad \mathbf{x}_i \leftarrow \mathbf{x}_{i-1} + \alpha_{i-1}\mathbf{p}_{i-1}$
10: $\quad\quad \mathbf{r}_i \leftarrow \mathbf{r}_{i-1} - \alpha_{i-1}\mathbf{s}_{i-1}$
11: $\quad\quad \mathbf{u}_i \leftarrow \mathbf{M}^{-1}\mathbf{r}_i$
12: $\quad\quad$ **for** $j = \max(0, i-k), \cdots, i-1$ **do**
13: $\quad\quad\quad \beta_{i,j} \leftarrow \frac{-1}{\eta_j} <\mathbf{A}\mathbf{u}_i, \mathbf{s}_j>$ $\qquad\qquad$ ▷ Allreduce(sum) of min(i,k) doubles
14: $\quad\quad \mathbf{p}_i \leftarrow \mathbf{u}_i + \sum_{j=\max(0,i-k)}^{i-1} \beta_{i,j}\mathbf{p}_j$
15: $\quad\quad \mathbf{s}_i = \mathbf{A}\mathbf{p}_i$
16: $\quad\quad \gamma_i \leftarrow <\mathbf{u}_i, \mathbf{s}_i>, \eta_i \leftarrow <\mathbf{s}_i, \mathbf{s}_i>$ $\qquad\qquad$ ▷ Allreduce(sum) of two doubles
17: $\quad\quad \alpha_i \leftarrow \frac{\gamma_i}{\eta_i}$
18: $\quad$ **return** $\mathbf{x}_i$

---

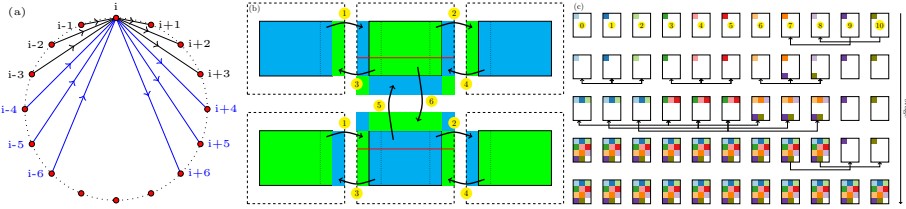

Fig. 4: Algorithms for three key MPI operations: (a) is the ring-k algorithm with k radix for all-to-all personalized communication generalized from ring alltoallv algorithm, (b) is the halo exchange algorithm, and (c) is the recursive-k algorithm with k radix generalized from the recursive doubling algorithm.





Table 3: A three-dimensional grid for assessing the communication of the atmospheric model. $\Delta x$ and $\Delta y$ are given as if this grid is a uniform global longitude-latitude grid. In fact, this grid resembles the grid of a regional spectral atmospheric model or the uniform longitude-latitude grid used by some global models.

| nx | ny | nz | $\Delta x$ | $\Delta y$ | grid points |
|---|---|---|---|---|---|
| 28800 | 14400 | 256 | 0.0125° | 0.0125° | > 100 billion |
| memory size | | | max processes | | |
| > 800 GB per double field | | | 3686400 for a 2D partition | | |

## 4 Experimental Results

### 4.1 Experiment Design

In the next decade, it is estimated the resolution of global NWP model will approach kilometre-scale and the HPC will move towards exascale. What would the performance of a global NWP model with a very high resolution on exascale HPC be? In this paper, we are especially interested in the strong scaling of an atmospheric model, that is, how does the atmospheric model with fixed resolution (such as the one presented in Table 3) behave as the number of processes increases? Table 3 presents a summary of the three-dimensional grid for assessing the communication of the kilometre-scale atmospheric model. The number of grid points of this grid is beyond 100 billion, and one field of double precision variable for this grid requires more than 800 gigabytes of memory. Only with such a large grid, is it possible to perform a 2D domain decomposition for a spectral model with more than one million processes so that modelling the communication of the atmospheric model at exascale HPC become possible.

Besides the topology and its configuration, the routing algorithm, and the collective MPI algorithm; the bandwidth and the latency of the interconnect network of an HPC system have a great impact on the performance of communications. First, we simulate the transposition for the spectral transform method in the simulator for three topologies (torus-M, fattree-M, and dragonfly-MM in Table 1), three configurations of dragonfly topology (dragonfly-MM, dragonfly-SL, and dragonfly-LS in Table 1), three routing algorithms (minimal, valiant, and ugal), and three alltoallv algorithms (Table 2). In addition, we compare the simulations of the transposition for the spectral transform method between four interconnect bandwidths ($10^0$, $10^1$, $10^2$, and $10^3$ GB/s) and between four interconnect latencies ($10^1$, $10^2$, $10^3$, and $10^4$ ns). After a thorough investigation of the transposition for the spectral transform method, we test



the halo exchange for the SL method with different halo widths (3, 10, 20, and 30 grid points), three topologies (torus-L, fattree-L, dragonfly-ML in Table 1), and three routing algorithms (minimal, valiant, and ugal). Finally, the allreduce operation in Krylov subspace methods for the SI method is evaluated on different topologies (torus-L, fattree-M, dragonfly-ML in Table 1), and the statistics of the optimal radix of recursive-k algorithms for allreduce operations are presented.

## 4.2 Transposition for the Spectral Transform Method

Fig. 5a shows that the communication times for the burst, bruck, ring-1, and ring-4 algorithms decrease as the number of MPI processes increases. The ring-1 and ring-4 algorithms are almost identical for less than $5 \times 10^4$ MPI processes, but ring-4 performs better than ring-1 for more than $10^5$ MPI processes. The burst and bruck algorithms perform worse than the ring-k algorithm. The SST/macro simulator cannot simulate the burst algorithm for more than $2 \times 10^4$ MPI processes because the burst messages result in huge events and large memory footprint. The communication time of the bruck algorithm is significantly larger than that of the ring-k algorithm for less than $10^5$ MPI processes; however, for a greater number of processes, it is better than the ring-1 algorithm since the bruck algorithm is targeted for small messages, and the more processes, the smaller message for a fixed sized problem. The performance of these alltoallv algorithms is confirmed by actually running the skeleton program of transposition for the spectral transform method with $10^4$ MPI processes on the research cluster of Météo France (Beaufix), which shows that the ring-4 algorithm is even better than the INTEL native MPI_Alltoallv function (Fig. 6).

The differences in the communication times of the transpositions between the topology torus-M, fattree-M, and dragonfly-MM can be an order of magnitude (Fig. 5b). Messages have to travel a long distance in the topology torus-M which is a 3D torus, so its communication time is the largest. The best performance of the topology fattree-M can be attributed to its non-blocking D-mod-k routing algorithm, but its communication time gradually increases as the number of MPI processes increases beyond $10^4$. The performance of topology dragonfly-MM is between that of torus-M and fattree-M (Fig. 5b), it can achieve a better performance by tuning the configuration of the dragonfly topology (Fig. 5c). By comparing Fig. 5b and Fig. 5c, we can see that the topologies of dragonfly-SL and dragonfly-LS are still not as good as the



fattree-M, but their performance is very close to that of fattree-M and they lose less scalability than fattree-M for more than $5 \times 10^4$ MPI processes.

The differences in communication time of the transpositions between the routing algorithms of minimal, valiant and ugal are also an order of magnitude (Fig. 5d), which indicates that the impact of routing algorithm on communication is significant. The valiant routing algorithm performs the best, but the communication time begins to increase when the number of MPI processes is larger than $3 \times 10^4$. The ugal routing algorithm performs the worst, and the performance of minimal routing algorithm is in between that of valiant and ugal routing algorithms. The valiant routing algorithm has the longest path for messages from the source to the destination with a randomly chosen intermediate node; thus, theoretically, its communication time is larger. On the contrary, the minimal routing algorithm that moves the messages using the shortest path from the source to the destination has the smallest communication time. The congestion between processes in Fig. 7 shows that the valiant routing algorithm for the dragonfly-MM topology (Fig. 7b) and the minimal routing algorithm for the dragonfly-SL topology (Fig. 7d) are less congested and have a more uniform congestion, the minimal routing algorithm for the dragonfly-MM topology is moderately congested, but its congestion is not uniform (Fig. 7a), the congestion of the ugal routing algorithm for the dragonfly-MM topology is large and highly non-uniform (Fig. 7c). These congestions in Fig. 7 are consistent with the communication times in Fig. 5c and Fig. 5d, that is, the more uniform congestion, the lower communication time because the latter is determined by the longest delay event and uniform congestion can avoid the hotspot of the congestion with the longest delay event. Fig. 8 confirms this that a high percentage of delay events has a delay time of less than 30 us using the valiant routing algorithm for the dragonfly-MM topology and the minimal routing algorithm for the dragonfly-SL topology; however the minimal routing algorithm for the dragonfly-MM topology has a significant percentage of events that delays by more than 50 us, especially there are a large number of events delayed by more than 100 us using the ugal routing algorithm for the dragonfly-MM topology. Thus, the configuration of the interconnect network and the design of its routing algorithm should make the congestion as uniform as possible if congestion is inevitable.

Although the communication time with a bandwidth of $10^0$ GB/s is apparently separated from those with bandwidths of $10^1$, $10^2$, and $10^3$ GB/s, the curves describing the communication times with bandwidths of $10^1$, $10^2$, and $10^3$ GB/s overlap (Fig. 5e). The communication times



with latencies of $10^1$ and $10^2$ ns are almost identical; that with a latency of $10^3$ $(10^4)$ ns is

slightly (apparently) different from those with latencies of $10^1$ and $10^2$ ns (Fig. 5f). Equation

(1) indicates that the communication time stops decreasing only when $\alpha$ $(\beta)$ approaches zero and

$\beta$ $(\alpha)$ is constant. Neither $\alpha$ in Fig. 5e nor $\beta$ in Fig. 5f approaches zero, but the communication

time stops decreasing. The inability of the analytical model (1) to explain this suggests that

other dominant factors such as congestion contribute to the communication time. Latency

is the amount of time required to travel the path from one location to another. Bandwidth

determines how many data per second can be moved in parallel along that path, and limits the

maximum number of packets travelling in parallel. Because both $\alpha$ and $\beta$ are greater than zero,

congestion occurs when data arrives at a network interface at a rate faster than the media can

service; when this occurs, packets must be placed in a queue to wait until earlier packets have

been serviced. The longer the wait, the longer the delay and communication time. Fig. 8b and

Fig. 8c show the distributions of the delay caused by congestion for different bandwidths and

different latencies, respectively. In Fig. 8b, the distributions of the delay for bandwidths of $10^1$,

$10^2$, and $10^3$ GB/s are almost identical, which explains their overlapped communication times

in Fig. 5e; and the distribution of the delay for a bandwidth of $10^0$ GB/s is distinct from the

rest since near 20 percent of events are delayed by less than 10 us but a significant percentage

of events are delayed more than 100 us, which accounts for its largest communication time in

Fig. 5e. In Fig. 8c, the distributions of the delay for latencies of $10^1$ and $10^2$ ns are the same;

the distributions of the delay for a latency of $10^3$ ns is slightly different from the formers; but

the distributions of the delay for a latency of $10^4$ ns has a large percentage of events in the

right tail which resulted in the longest communication time; these are consistent with their

communication times in Fig. 5f.

In summary, the alltoallv algorithm, the topology and its configuration, the routing al-

gorithm, the bandwidth, and the latency have great impacts on the communication time of

transpositions. In addition, the communication time of transpositions decreases as the number

of MPI processes increases in most cases; however, this strong scalability is not applicable for

the fattree-M topology (the red line in Fig. 5b), the dragonfly-SL and dragonfly-LS topologies

(red and black lines in Fig. 5c), and the valiant routing algorithm (the red line in Fig. 5d) when

the number of MPI processes is large. Thus, the topology of the interconnect network and its

routing algorithm have a great impact on the scalability of transpositions for the spectral trans-

form method. Since the transposition for spectral transform method is a multiple simultaneous



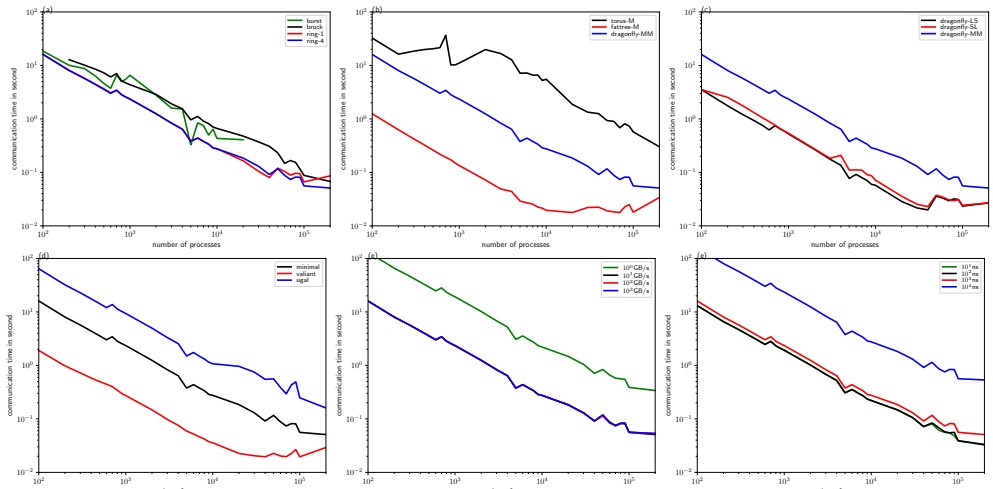

Fig. 5: Communication times of transposition for (a) alltoallv algorithms, (b) topologies, (c) configurations of the dragonfly topology, (d) routing algorithms for the dragonfly topology, (e) bandwidth, and (f) latency.

all-to-all personalized communication, congestion has a great impact on its performance.

### 4.3 Halo Exchange for the Semi-Lagrangian Method

The most common application of the wide halo exchange is the SL method. For the resolution of $0.0125°$ in Table 3 and a time step of 30 seconds, the departure is approximately 5 grid points away from its arrival if the maximum wind speed is 200 m/s; therefore, the width of the halo is at least 7 grid points using the ECMWF quasi-cubic scheme (Ritchie, 1995); there are more grid points if a higher order scheme such as the SLICE-3D (Zerroukat and Allen, 2012)

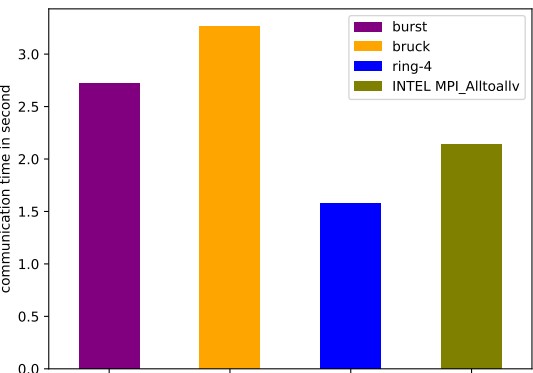

Fig. 6: Actual communication time of transposition for the spectral transform method with $10^4$ MPI processes run on beaufix cluster in Météo France.





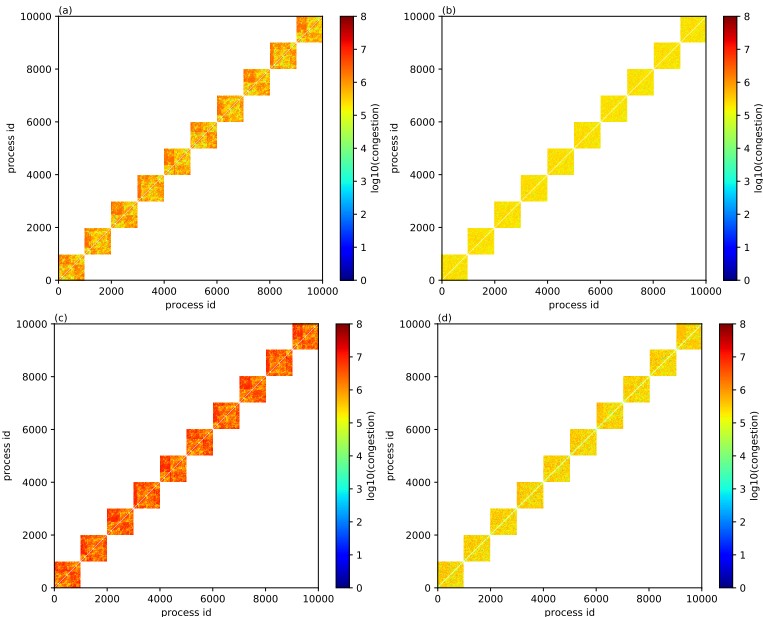

Fig. 7: Congestion of transposition using (a) minimal routing algorithm for the dragonfly-MM topology, (b) valiant routing algorithm for the dragonfly-MM topology, (c) ugal routing algorithm for the dragonfly-MM topology, and (d) minimal routing algorithm for the dragonfly-SL topology.

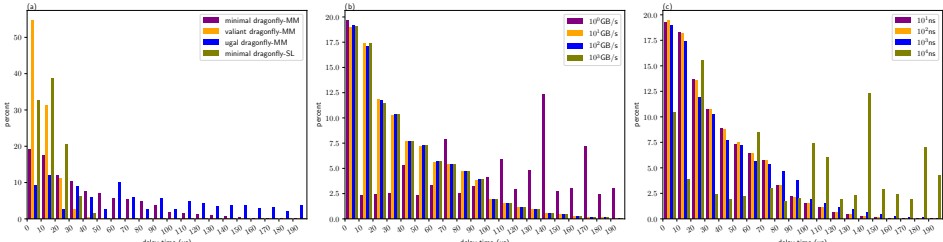

Fig. 8: Distribution of delayed events of transposition for the spectral transform method with $10^4$ MPI processes using (a) different routing algorithms and topology configurations, (b) different bandwidths, and (c) different latencies, simulated by SST/macro.

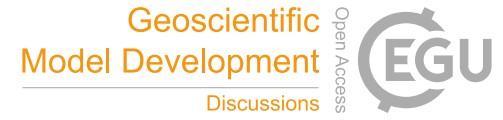



is used. In Fig. 9a, the communication time of the halo exchange decreases more slowly as the number of processes increases than that of transposition for the spectral transform method. This is because the message size decreases more slowly than that of transposition owing to the fixed width of the halo (figure omitted). If the communication time of the transposition (halo exchange) continues its decreasing (increasing) trend in Fig. 9a, they meet at certain number of MPI processes; then, the communication time of the halo exchange is larger than that of the transposition. In addition, it can be seen that the wider the halo, the longer the communication time. The halo exchange of a thin halo of 3 grid points, for such as the 6th order central difference $F'_i = \frac{-F_{i-3} + 9F_{i-2} - 45F_{i-1} + 45F_{i+1} - 9F_{i+2} + F_{i+3}}{60\Delta}$ (the red line in Fig. 9a), is significantly faster than that of wide halo for the SL method (green and blue lines in Fig. 9a). Thus, the efficiency of the SL method is counteracted by the overhead of the wide halo exchange where the width of the halo is determined by the maximum wind speed. Wide halo exchange for the SL method is expensive at exascale, especially for the atmospheric chemistry models where a large number of tracers need to be transported. On-demand exchange is a way to reduce the communication of halo exchange for the SL method, and will be investigated in a future study.

Significant differences in the communication times of the wide halo exchange of 20 grid points for topology torus-L, fattree-L, and dragonfly-ML are shown in Fig. 9b. It can be seen that topology torus-L performs the worst, fattree-L is the best, and the performance of dragonfly-ML is between that of torus-L and fattree-L. The communication time of the wide halo exchange of 20 grid points for the topology tour-L abruptly increases at approximately $10^3$ MPI processes, and then gradually decreases when the number of MPI tasks becomes larger than $3 \times 10^3$ MPI processes. The impact of the routing algorithm on the communication time of the wide halo exchange of 20 grid points (Fig. 9c) is the same as on that of transposition (Fig. 5d): the routing algorithm valiant performs the best, the routing algorithm ugal performs the worst, and the routing algorithm minimal is between valiant and ugal.

### 4.4   Allreduce in Krylov Subspace Methods for the Semi-Implicit Method

If, in average, the GCR with a restart number $k = 3$ is convergent with $N = 25$ iterations, the number of allreduce calls is $2 \times N = 50$. The black and blue lines are the communication times of 50 allreduce operations using MPI_Allreduce and the recursive-k algorithm, respectively;



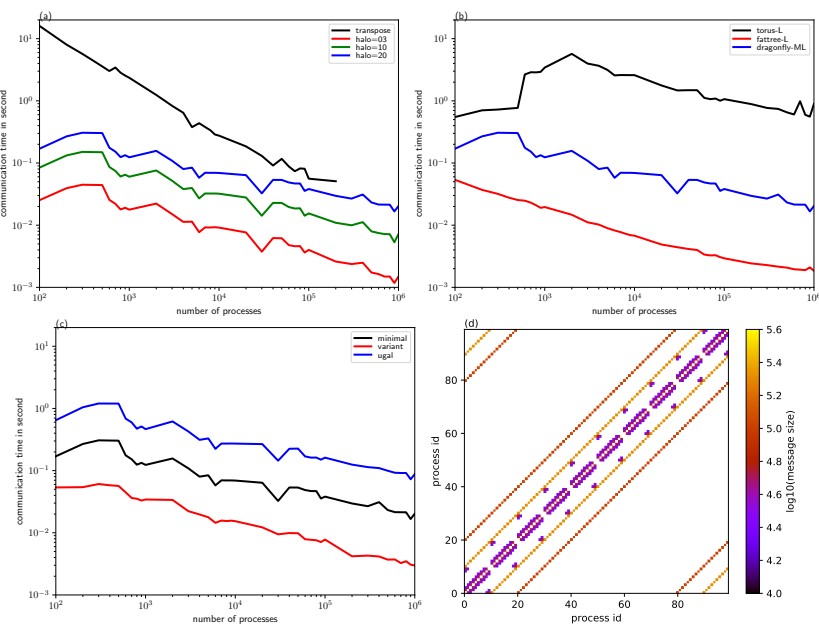

Fig. 9: (a) is the communication times of the halo exchange with a halo of 3 (red line), 10 (green line), and 20 (blue line) grid points, and the communication time of transposition for the spectral transform method is shown for comparison (black line). (b) is the communication times of the halo exchange with a halo of 20 grid points for the topology of torus-L (black line), fattree-L (red line), and dragonfly-ML (blue line). (c) is the communication times of the halo exchange with a halo of 20 grid points for the routing algorithm of minimal (black line), valiant (red line), and ugal (blue line). (d) illustrates the communication pattern of the halo exchange with a wide halo.





that is, the estimated communication time of one single GCR call (Fig. 10a). Contrary to that
of transposition, the communication time of GCR increases as the number of MPI processes
increases. Following the trend, the communication of a single GCR call may be similar to or
even larger than that of a single transposition when the number of MPI processes approaches
to or is beyond one million. Although it is believed that the spectral method does not scale
well owing to its time-consuming transposition, it does not suffer from this expensive allreduce
operation for the SI method because of its mathematical advantage that spherical harmonics are
the eigenfunctions of Helmholtz operators. In this sense, a grid-point model with the SI method
in which the three-dimensional Helmholtz equation is solved by Krylov subspace methods may
also not scale well at exascale unless the overhead of allreduce communication can be mitigated
by overlapping it with computation (Sanan et al., 2016).

Fig. 10b shows the communication times of allreduce operations using the recursive-k algo-
rithm on the topologies of torus-L, fattree-L, and dragonfly-ML. The impact of topology on the
communication performance of allreduce operations is obvious. The topology of torus-L has the
best performance, but is similar to that of dragonfly-ML for more than $5 \times 10^5$ MPI processes;
and fattree-L has the worst performance. However, the impact of three routing algorithms
(minima, valiant, and ugal) for the dragonfly-ML topology has a negligible impact on the com-
munication performance of allreduce operations (figure omitted); this may be because of the
tiny messages (only 3 doubles for the restart number $k = 3$) communicated by the allreduce
operation.

One advantage of the recursive-k algorithm of the allreduce operation is that the radix k
can be selected to reduce the stages of communication by making full use of the bandwidth
of the underlying interconnect network. We repeat the experiment, whose configuration is
as that of the blue line in Fig. 10a, for the proper radix $k \in [2, 32]$, and the optimal radix
is that with the lowest communication time for a given number of MPI processes. For each
number of MPI processes, there is an optimal radix. The statistics of all the optimal radices are
shown in Fig. 10c. It can be seen that the minimum and maximum optimal radices are 5 and
32, respectively. Thus, the recursive doubling algorithm that is equivalent to the recursive-k
algorithm with radix k=2 is not efficient since the optimal radix is at least 5. The median
number of optimal radices is approximately 21, and the mean number is less than but very
close to the median number. We cannot derive an analytic formula for the optimal radix since
modelling the congestion is difficult in an analytic model. However, for a given resolution of

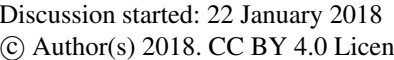



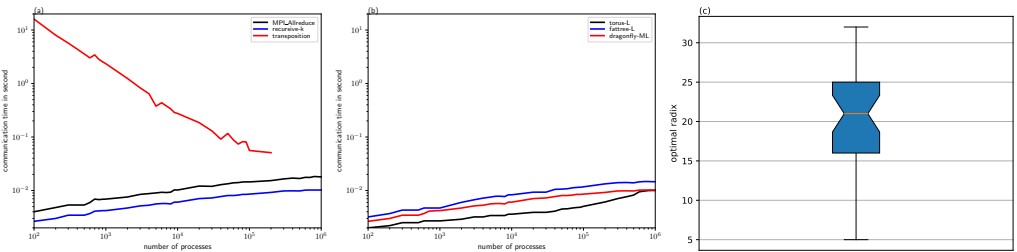

Fig. 10: (a) is the communication times of the allreduce operation using the MPI_Allreduce
(black line) and the recursive-k algorithm (blue line), and the communication time of trans-
position for the spectral transform method is shown for comparison (red line). (b) is the
communication times of the allreduce operation using the recursive-k algorithm for the topol-
ogy torus-L (black line), fattree-L (blue line), and dragonfly-ML (red line). (c) is the statistics
of the optimal radices for the recursive-k algorithm.

NWP model and a given HPC system, fortunately, the number of processes, bandwidth, and
latency are fixed; thus, it is easy to perform experiments to obtain the optimal radix.

# 5   Conclusion and Discussion

This work shows that it is possible to make simulations of the MPI patterns commonly used in
NWP models using very large numbers of MPI tasks. This enables the possibility to examine
and compare the impact of different factors such as latency, bandwidth, routing and network
topology on response time. We have provided an assessment of the performance and scalability
of three key MPI operations in an atmospheric model at exascale by simulating their skeleton
programs on an SST/macro simulator. After optimization of the memory and efficiency of
the SST/macro simulator and construction of the skeleton programs, a series of experiments
was carried out to investigate the impacts of the collective algorithm, the topology and its
configuration, the routing algorithm, the bandwidth, and the latency on the performance and
scalability of transposition, halo exchange, and allreduce operations. The experimental results
show that:

1. The collective algorithm is extremely important for the performance and scalability of
   key MPI operations in the atmospheric model at exascale because a good algorithm can
   make full use of the bandwidth and reduce the stages of communication. The generalized
   ring-k algorithm for the alltoallv operation and the generalized recursive-k algorithm for
   the allreduce operation proposed herein perform the best.





2. Topology, its configuration, and the routing algorithm have a considerable impact on the performance and scalability of communications. The fattree topology usually performs the best, but its scalability becomes weak with a large number of MPI processes. The dragonfly topology balances the performance and scalability well, and can maintain almost the same scalability with a large number of MPI processes. The configurations of the dragonfly topology indicate that a proper configuration can be used to avoid the hotspots of congestion and lead to good performance. The minimal routing algorithm is intuitive and performs well. However, the valiant routing algorithm (which randomly chooses an intermediate node to uniformly disperse the communication over the network to avoid the hotspot/bottleneck of congestion) performs much better for heavy congestion.

3. Although they have an important impact on communication, bandwidth and latency cannot be infinitely grown and reduced owing to the limitation of hardware, respectively. Thus, it is important to design innovative algorithms to make full use of the bandwidth and to reduce the effect of latency.

4. It is generally believed that the transposition for the spectral transform method, which is a multiple simultaneous all-to-all personalized communication, poses a great challenge to the scalability of the spectral model. This work shows that the scalability of the spectral model is still acceptable in terms of transposition. However, the wide halo exchange for the Semi-Lagrangian method and the allreduce operation in the GCR iterative solver for the Semi-Implicit method, both of which are often adopted by the grid-point model, also suffer the stringent challenge of scalability at exascale.

In summary, both software (algorithms) and hardware (characteristics and configuration) are of great importance to the performance and scalability of the atmospheric model at exascale. The software and hardware must be co-designed to address the challenge of the atmospheric model for exascale computing.

As shown previously, the communications of the wide halo exchange for the Semi-Lagrangian method and the allreduce operation in the GCR iterative solver for the Semi-Implicit method are expensive at exascale. The on-demand halo exchange for the Semi-Lagrangian and the pipeline technique to overlap the communication with the computation for the GCR iterative solver are not researched in this study and should be investigated. All the computed nodes in this work only contain one single-core CPU, which is good for assessing the communication of



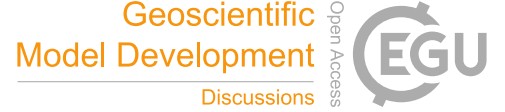

the interconnect network; however, it is now very common for one CPU with multi-cores or even many-cores. Multiple MPI processes per node may be good for the local pattern communication such as thin halo exchange since the shared memory communication is used, but may result in heavy congestion in the network interface controller for all-to-all communication. The more MPI processes, the less computation per node without limitation if there is only one single-core CPU per node, thus, computation is not considered in this paper. However, the bandwidth of memory limits the performance and scalability of computation for multi-core or many-core systems. The assessment of computation currently underway and a detailed paper will be presented separately; the purpose of this subsequent study is to model the time response of a time step of a model such as the regional model (AROME) used by Météo-France.

## Code Availability

The code of the SST/macro simulator is publicly available at https://github.com/sstsimulator/sst-macro. The skeleton programs, scripts, and our modified version of SST/macro 7.1.0 for the simulations presented the paper are available at https://doi.org/10.5281/zenodo.1066934.

## Competing Interests

The authors declared no competing interests.

## Acknowledgements

This work was supported by Centre National de Recherches Météorologiques, Météo France within the ESCAPE (Energy-efficient Scalable Algorithms for Weather Prediction at Exascale) project.

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
