# Peer review of "Simulation of the Performance and Scalability of MPI Communications of Atmospheric Models running on Exascale Supercomputers"

_Geoscientific Model Development, 2017_

## Referee Comment (RC1) · Anonymous Referee #1 · 9 Feb 2018

The authors present work of simulated scaling analysis for different communication algorithms commonly used in atmospheric models using a skeleton codes and a simulation package to examine the scaling performance on possible future supercomputers. This represents significant new information on how these algorithms may perform and is likely to be of interest to the community. The methods used are well described and appear robust. Some of the assumptions made about future architectures, in effect single CPU core nodes, are unlikely to be entirely valid. Whilst these are made the entirely reasonable purpose of make the simulations tractable, they may weaken some of

the conclusions. For example, almost all CPU base supercomputer are multi-core and multi-socket nodes which then have significant network hierarchy. Moreover, many of the largest machines in the top 500 list have non-CPU architectures such as GPUs and Xeon Phi. These have more complex hierarchies and are unlikely to, or even cannot be, programmed with a single MPI rank bound to single "core". Whilst the authors don't hide this, this is not discussed in the conclusions. Most of the results are presented in the form of graphs. Unfortunately, they are simply too small and it not possible to read the legends, axis labels etc. This makes it difficult to judge the quality of the results and the inferences drawn. These should be reproduced to appear much larger. Moreover, it would appear (although hard to be sure) that some of the plots have number of processors as the x-axis. This is a discrete variable and so line graphs should not be used, a bar chart may be appropriate. Whilst it may be common practice to present scaling data in this way, it is still wrong. This paper has the potential to become an interesting and significant work, but not in its current form. Once some revisions have been made it should be review again. In particular, there are three changes which are necessary. i) The plots must be made bigger so they are legible ii) Plots against discrete variables shouldn't be line graphs iii) The authors should comment on and discuss what conclusions can be drawn from simulations of single core nodes for more complex node architectures and the consequent differences to communication patterns.
* * *

---

## Author Comment (AC1) · 19 Mar 2018

Dear reviewer,

Thank you very much for your comments. The followings are our responses. Please also find the revised manuscript in the supplement to this comment.

Best regards,

Yongjun ZHENG and Philippe MARGUINAUD

[Figure]
* * *
Reviewer #1:

The authors present work of simulated scaling analysis for different communication algorithms commonly used in atmospheric models using a skeleton codes and a simulation package to examine the scaling performance on possible future supercomputers. This represents significant new information on how these algorithms may perform and is likely to be of interest to the community. The methods used are well described and appear robust.

Thank you very much for your careful comments.

Some of the assumptions made about future architectures, in effect single CPU core nodes, are unlikely to be entirely valid. Whilst these are made the entirely reasonable purpose of make the simulations tractable, they may weaken some of the conclusions. For example, almost all CPU base supercomputer are multi-core and multi-socket nodes which then have significant network hierarchy. Moreover, many of the largest machines in the top 500 list have non-CPU architectures such as GPUs and Xeon Phi. These have more complex hierarchies and are unlikely to, or even cannot be, programmed with a single MPI rank bound to single "core". Whilst the authors don't hide this, this is not discussed in the conclusions.

The main purpose of this study is to analyse the performance and scalability of communications over an interconnect network between nodes. Thus, single CPU core per node is adopted; because this not only makes the simulations tractable, but also eliminates the intra-node communications, which in turn makes it easy to draw robust conclusions for the inter-node communications without the complicated hierarchical

network. But we totally agree with the reviewer that the architectures of current and futures supercomputers are multi-core and multi-socket nodes, even non-CPU architectures; intra-node communications significantly distinguish from inter-node communications. For example, some MPI implementations implement the intra-node communication using the shared memory communication mechanism for multi-core and multi-socket nodes, or using proprietary inter-processor networks and API for non-CPU architectures. However, an MPI rank can be bound to any core for multi-core and multi-socket nodes; and an MPI rank can be bound to any processor/co-processor for MIC architectures such as Xeon Phi; with CUDA-aware MPI, an MPI rank can be bound to a CPU core but can communicated with GPUs for GPU architectures. Because A multi-core node behaves more or less like a more powerful single core node when the OpenMP is used for the intra-node parallelization, the assumption of an MPI rank bound to a single core should apply to the complex hierarchical system. We have added a discussion for more complex hierarchical architectures in the conclusions. Please refer to the conclusions section in the revised manuscript.

Most of the results are presented in the form of graphs. Unfortunately, they are simply too small and it not possible to read the legends, axis labels etc. This makes it difficult to judge the quality of the results and the inferences drawn. These should be reproduced to appear much larger.

Thank you very much for pointing out the regibility of some figures. We have reproduced most the figures so that they are legible, especially for the legends and axis labels.

Moreover, it would appear (although hard to be sure) that some of the plots have number of processors as the x-axis. This is a discrete variable and so line graphs should not be used, a bar chart may be appropriate. Whilst it may be common practice to present scaling data in this way, it is still wrong. This paper has the potential to become an interesting and significant work, but not in its current form.

Fig.5, Fig.9a-c, and Fig.10a-b have number of processes as the x-axis which is a discrete variable. In the revised manuscript, we have added one statement (lines 405-408) about the discrete values adopted in this study. We have tried to change the line plots to a bar chart, but it is not as clear as a line to demonstrate the trend of communications times, which varies as the number of processes increases. But we changed the lines to the lines with markers which indicates the number of processes, and added explanations in the captions of the figures. Thank you again for your careful comments.

Once some revisions have been made it should be review again. In particular, there are three changes which are necessary.

i) The plots must be made bigger so they are legible

We have changed the Fig.3, Fig.4, especially, Fig.5, Fig.8, Fig.9, and Fig.10 so that they are legible now. Please refer to the revised manuscript.

ii) Plots against discrete variables shouldn't be line graphs
As mentioned above, we have changed the Fig.5, Fig.9a-c, and Fig.10a-b. Please refer to the revised manuscript.

iii) The authors should comment on and discuss what conclusions can be drawn from simulations of single core nodes for more complex node architectures and the consequent differences to communication patterns.

As mentioned above, the binding of an MPI rank is possible for non-CPU architectures; thus, the conclusions for inter-node communications could be generalized to more complex node architectures. As we already discussed, the intra-node communications significantly distinguish from inter-node communications, and multiple MPI processes per node in complex node architectures may result in congestion in the network interface controller for inter-node communication. The congestion can be mitigated even eliminated if more network interface controllers per node or a network interface controller with multi-ports (such as a mini-switch) in a node. From this point of view, our conclusion should still be valid for this complex hierarchical architectures, but the scalability might be affected. We agreed with the reviewer that a discussion should be included and we have added a statement about this. Please refer to the conclusions section in the revised manuscript.

Please also note the supplement to this comment:
https://www.geosci-model-dev-discuss.net/gmd-2017-301/gmd-2017-301-AC1-supplement.pdf

––––––––––––––––––––––––––––

**Supplement:**

[revised manuscript text omitted]

---

## Referee Comment (RC2) · Anonymous Referee #1 · 23 Apr 2018

The authors have made appropriate modifications to the paper and discussed what they have changed and what they have not in their reply. The main issue was the legibility of the figures which has now been improved in this revision. There are some minor typographical errors in the new section on likely node architectures of future systems. Page 29, line 639 "futures"–> "future. Line 645, the "A" is capitalised in the middle of a sentence. Line 651 "mitigated even eliminated if" –> "mitigated or even eliminated, if", this sentence then somewhat loses its way, I think it should be something like "network interfaces added". These errors should be addressed, but they should not

require subsequent review.

The paper reports on the simulation of scaling of three important algorithms for weather and climate codes to a very large degree of parallelism. The methods used are well described and sufficiently robust to support the conclusions. Where the are limitations to the methods the authors have appropriately drawn attention to them. This is a very relevant paper which is likely to be of interest to the community which advances the knowledge of the performance of algorithms on future super-computer architectures. As such, it is an important work for model developers and should now be published.

―――――――――――――――――

---

## Author Comment (AC2) · 24 Apr 2018

Dear reviewer,

Thank you so much for pointing the typographical errors out to us. We have corrrected the typographical errors, and clarified the lost sentence as "The congestion can be mitigated or even eliminated, if each node has more network interface controllers (NICs) or a network interface controller with multi-ports (as a mini-switch)." These changes will be in the final revision of the manuscript.

[Figure]

Best regards,

Yongjun ZHENG (on behalf of all Co-Authors)
* * *

---

## Referee Comment (RC3) · Anonymous Referee #2 · 4 May 2018

The article presents an important aspect often ignored in NWP model development. It studies the impact of network topology, not only for one particular algorithm, but for multiple representative algorithms found in NWP models. It illustrates that the choice of equivalent but different numerical algorithms may well depend on the available network layout. In this case a semi-Lagrangian approach using nearest-neighbour communication for wide halo-exchange is studied. Further, a spectral transform method is studied consisting of large distributed matrix transpositions, and finally a Krylov solver consisting of multiple AllReduce operations. The results are presented in a detailed yet clear manner.

I have attached a edited PDF of the original article containing comments and suggestions. If these are addressed, I am happy to see the article published.

For clarity, I will report the major comments and questions below (besides being present already in the attached PDF).

1) Throughout the article the term "radix" is used. It would be good to formulate a definition of it in this paper's context.

2) Line 135: I recommend following more representative citation instead of "Kuhnlein et al., 2017": Smolarkiewicz et al., 2016: A finite-volume module for simulating global all-scale atmospheric flows, J. Comput. Phys., 314, pp. 287-304, doi:10.1016/j.jcp.2016.03.015

3) Line 363: It's worth noting that this regularity is only possible for structured grids. Even then there are differences between regular and reduced grids. Unstructured grids would not have a preferred x or y sweeping, and communication must be done in a single sweep. Does the following analysis still hold in this case?

4) Line 603, whole paragraph: Can MPI tasks be carefully pinned to cores using knowledge of the domain decomposition to reduce congestion?

5) Line 639: "However, the bandwidth of memory limits the performance and scalability of computation for multi-core or many-core systems". This statement seems taken without reasoning. Surely this cannot apply to any algorithm. Could the authors elaborate?

6) Acknowledgements: The Horizon 202 program ESCAPE acknowledgement has more strict rules on how to acknowledge (e.g. mention of EU and program number). I recommend asking the project manager for details.

Please also note the supplement to this comment:

https://www.geosci-model-dev-discuss.net/gmd-2017-301/gmd-2017-301-RC3-supplement.pdf

**Supplement:**

[revised manuscript text omitted]

---

## Author Comment (AC3) · 11 May 2018

Dear reviewer,

Thank you very much for your comments. The followings are our responses. Please also find the revised manuscript in the supplement.

Best regards,

Yongjun ZHENG and Philippe MARGUINAUD

[Figure]
* * *
Reviewer #2:

The article presents an important aspect often ignored in NWP model development. It studies the impact of network topology, not only for one particular algorithm, but for multiple representative algorithms found in NWP models. It illustrates that the choice of equivalent but different numerical algorithms may well depend on the available network layout. In this case a semi-Lagrangian approach using nearest-neighbour communication for wide halo-exchange is studied. Further, a spectral transform method is studied consisting of large distributed matrix transpositions, and finally a Krylov solver consisting of multiple AllReduce operations. The results are presented in a detailed yet clear manner.

I have attached a edited PDF of the original article containing comments and suggestions. If these are addressed, I am happy to see the article published.

We really appreciate your comments and your efforts to edit the original manuscript.

For clarity, I will report the major comments and questions below (besides being present already in the attached PDF).

1) Throughout the article the term 'radix" is used. It would be good to formulate a definition of it in this paper's context.

After searched the term 'radix" in the manuscript, we found there are mainly three parts that uses the term 'radix":

1.  In Table 1 in page 10, the last column 'radix" is related to the number of ports of

a switch. We found this radix is never referenced in this paper, so it is removed from the Table 1 to avoid the confusion with the following two usages.

2. In Table 2 in page 14, in the description of the ring-k algorithm for a spectral transposition, the 'radix k" represents the number of processes to (from) which a process sends (receives) messages. Thus, the 'radix k" is self-explained.

3. The remainning usages of the term 'radix" are for the recursive-k algorithm for the allreduce operation. The 'radix k" represents the number of processes involved in a sub-reduce operation of the resursive-k algorithm. Since this is not obvious, we added the definition of 'radix k" and made some changes so that the description of the recursive-k algorithm is more accurate. Please refer to the revised manuscript (lines 390-404).

2) Line 135: I recommend following more representative citation instead of "Kuhn-lein et al., 2017": Smolarkiewicz et al., 2016: A finite-volume module for simu-lating global all-scale atmospheric flows, J. Comput. Phys., 314, pp. 287-304, doi:10.1016/j.jcp.2016.03.015

The citation has be changed, please refer to line 136 in the revised manuscript.

3) Line 363: It's worth noting that this regularity is only possible for structured grids. Even then there are differences between regular and reduced grids. Unstructured grids would not have a preferred x or y sweeping, and communication must be done in a single sweep. Does the following analysis still hold in this case?

Yes, we agree with you that halo exchange for unstructured grids must be done in a sin-gle sweep. Two sweep method for a regular grid has the advantage that each process

only exchange messages with his TWO neighbors in the corresponding direction, less processes involved in a communication usually reduce the possibility of congestions; but two sweep method has an overhead time since the second sweep has to wait for the finish of the first sweep. One single sweep method can avoid the overhead time in the two sweep method; but its disadvantage is that each process need to communicate with its EIGHT neighbors simultanuously, this would increase the possibility of congestions. In short, these two methods should have be similar in term of communication times. We adapted the halo exchange skeleton program to the single sweep method, and compared the communication times between the two sweep method and the single sweep method. The result (see Fig. 1) for the halo exchange with a halo of 20 grid points show that the difference between two methods is minor. Thus, we believe the analyses in our paper are also held for unstructured grids.

4) Line 603, whole paragraph: Can MPI tasks be carefully pinned to cores using knowledge of the domain decomposition to reduce congestion?

The domain decomposition, the topology of the interconnet network, and the communication pattern all have an impact on the congestion. The halo exchange usually has a local communication pattern; thus, with the knowledges of the domain decomposition and the underlying topolopy, it is possible to pin MPI tasks to cores so that each process exchanges messages with the near processes to reduce congestion; for example, a regular 2D/3D domain decomposition is mapped to a 2D/3D torus network. The transpositon and allreduce operation are all-to-all communications, it is not easy, if not possible, to map MPI tasks to cores to reduce the congestion.

5) Line 639: "However, the bandwidth of memory limits the performance and scalability of computation for multi-core or many-core systems". This statement seems taken without reasoning. Surely this cannot apply to any algorithm. Could the authors elaborate?

Our intention of using the statement is to elicit the last sentence in the manuscript. Because this paper investigated the communication of atmospheric models using a single-core CPU per node, a singe-core CPU per node is good to assess the communication. But the architectures of current and future supercomputers are multi-core and multi-socket nodes, even non-CPU architectures. Because multi-core or many-core processors share a memory bus, it is possible for a memory-intensive application (such as an atmospheric model) to saturate the memory bus and result in degraded performances of all the computations running on that processor. Thus, our subsequent study will focus on the assessment of computations. We have clarified the last two sentences, please refer to the revised manuscript.

6) Acknowledgements: The Horizon 202 program ESCAPE acknowledgement has more strict rules on how to acknowledge (e.g. mention of EU and program number). I recommend asking the project manager for details.

Thank you very much for pointing out this to us. We have updated the acknowledgement to conform to the rules of the Horizon 2020 program ESCAPE.

In addiction, all the other sugestions presented in your attached PDF have been incorporated into the revised manuscript.

Please also note the supplement to this comment:
https://www.geosci-model-dev-discuss.net/gmd-2017-301/gmd-2017-301-AC3-supplement.pdf

[Figure]

2018.

**Fig. 1.** Communication times of wide halo exchange between the two sweep method and the single sweep method

**Supplement:**

[revised manuscript text omitted]

---

## Author Response (AR2)

Dear Dr Ham,

Thank you very much for your suggestions for improving the quality of our figures. We agreed with you that our figures should have a publication-quality to help the readers to take in the information from them. So we polished all the figures except for the Fig 1. Please refer to the following section and the revised version for the improvements we made for each figure. In addition, we corrected several spelling or grammar errors (see the last paragraphs in the following section).

We would like to express our sincere gratitude for your approval of the scope and scientific content of our very first manuscript, and your thoughtful suggestions (especially, for the quality of the English writing and the quality of the figures) that have helped improve this paper substantially.

Best regards,

Yongjun ZHENG and Philippe MARGUINAUD
* * *
It appears that your changes do substantially address the concerns of the reviewers, and in particular reviewer 1 has now indicated in private correspondence that he no longer needs to see the revised manuscript. However his point about font sizes in figures does still stand and still needs to be fixed. In particular, the figures are very inconsistent in this regard, with some being very good and others being awful. Specifically, you should be aiming for the text in the figures to be at least the size of footnote text for the main manuscript. With respect to the particular figures:

Fig 1. Very good. All figures should have fonts this size.

Thank you for thoroughly pointing out the problems of texts in the figures. Fig 1. and Fig 2. are adapted from other papers, Fig 3. and Fig 4. are produced using tikz, the remainders are produced using matplotlib. We have tried our best to make the fonts the same size. Finally, we found they do not have the same font completely due to the different sources or tools.

Fig 2. Much too small text. Rearrange into two rows so the figures can be big enough to read the text. Also, please make the fonts consistent. a, b, and c are currently all different, and "Compute Nodes", "Switch", and "Group" are all different. This just distracts the reader.

Thank you for the suggestion: we have rearranged the figures into two rows and these subfigures are legible now. Also we changed the texts of these three subfigures to have almost the same font in both face and size. Becausse the order of these subfigures is changed, the caption of Fig 2 and the references (between line 218 and 237) to them are also changed.

Fig 3. Text is slightly too small. Also font is not consistent with previous figures.

The font size has been increased by 1 point. And the bold face of the texts have been removed to be consistent with other figures.

Fig 4. The size of a is fine, though the fonts are once again internally inconsistent. b and c are much too small, both in font and in the diagrams. My eyes hurt trying to read them.

We rearranged the figures into two rows so that they become larger. In fact, these three subfigures use the same font. We believed that the slight inconsistency of the fonts is come from the different stretchings of these subfigures when combining them into a figure.

Fig 5. Good size, but why are the numbers in a different font from the text?

Due to the superscript, the numbers are displayed in a math formula mode that is why the numbers are different from the texts. Also, we found the different appearances of lowercase and uppercase texts look like that inconsistent fonts are used, indeed, the same font is used.

Fig 6. Axis label is a little small, otherwise good.

The axis label has been enlarged a bit.

Fig 7. Text is far too small. Images are also somewhat too small to decipher.

The axis labels and tick labels are enlarged. Also, removed the colorbars, axis labels, and tick labels between the subfigures to make more rooms for the images. Now the images are better than the old one.

Fig 8. Everything is too small, Basically unintelligible. Fonts are also inconsistent.

We rearranged the figure into one subfigure per row and significantly improved the quality. For the inconsistency in fonts, there is a difference between math mode and text mode as mentioned before. Also there are some slight differences between the axis labels, tick labels, and legend labels due to the internal setting of the plotting software, even we made them consistent as possible as we can.

Fig 9. Fonts possibly a little soma but not disastrous. Fonts are inconsistent.

The texts have been enlarged a bit. We have tried our best to make the fonts consistent.

Fig 10. Much too small text. Magnifying glass required.

The figure has been rearranged in two rows and enlarged the texts. Now its quality is quite better.

Corrected several spelling and grammar errors:

1. line 38: over than → over

2. line 86: 65536 → 65,536

3. line 128: causes → cause

4. line 219: taddthe → the

5. line 252: and the same nodes per switch

6. line 254: 390625 → 390,625

7. line 367: are due because of → are because of

8. line 393: decription → description

9. line 394: proccesses → processes

10. line 443, 448, 485, and 513: less than → fewer than

11. line 453: of Météo France (Beaufix) → (Beaufix) of Météo France

12. line 484: hotspot → hotspots

13. line 499: given a fixed message size

14. line 562: in average → on average

15. line 625: hotspot/bottleneck → hotspots

16. line 652: can communicated → can communicate

17. line 662: architecutres → architectures

18. line 673: the → in this

19. Table 1: the number of switches for dragonfly-SL and dragonfly-LS is changed from 15625 (a copy-paste mistake) to 78125

20. Acknowledgements: add a sentence for our sincere gratitude

[revised manuscript text omitted]